# Ribosome Display Technology: Applications in Disease Diagnosis and Control

**DOI:** 10.3390/antib9030028

**Published:** 2020-06-27

**Authors:** Adinarayana Kunamneni, Christian Ogaugwu, Steven Bradfute, Ravi Durvasula

**Affiliations:** 1Department of Medicine, Loyola University Medical Center, Chicago, IL 60153, USA; Ravi.Durvasula@lumc.edu; 2Department of Animal and Environmental Biology, Federal University Oye-Ekiti, Ekiti State 371010, Nigeria; christian.ogaugwu@fuoye.edu.ng; 3Center for Global Health, Department of Internal Medicine, University of New Mexico, Albuquerque, NM 87131, USA; sbradfute@salud.unm.edu

**Keywords:** ribosome display, antibodies, disease control, diagnostics

## Abstract

Antibody ribosome display remains one of the most successful in vitro selection technologies for antibodies fifteen years after it was developed. The unique possibility of direct generation of whole proteins, particularly single-chain antibody fragments (scFvs), has facilitated the establishment of this technology as one of the foremost antibody production methods. Ribosome display has become a vital tool for efficient and low-cost production of antibodies for diagnostics due to its advantageous ability to screen large libraries and generate binders of high affinity. The remarkable flexibility of this method enables its applicability to various platforms. This review focuses on the applications of ribosome display technology in biomedical and agricultural fields in the generation of recombinant scFvs for disease diagnostics and control.

## 1. Introduction

Antibodies have long been powerful tools for basic research, diagnostics, and treatment of diseases [1,2,3] and are currently the fastest-growing class of therapeutic molecules. Recombinant antibody (rAb) fragments are now emerging as promising alternatives to full-length monoclonal antibodies (mAbs) since they are smaller, retaining the targeting specificity of the whole mAbs, but can be produced more economically, are easily amenable to genetic manipulation, and possess other unique and superior properties that are advantageous in certain medical applications. Single-chain fragment variable (scFv) antibodies are one of the most widespread rAb formats, and they have been engineered for many applications [4,5].

Traditionally, mAbs were derived from rodents using hybridoma technology. Different display technologies have been used, more recently, to generate high-affinity, specific, and stable mAbs. Display technology is usually used to isolate the DNA or RNA encoding a selected protein sequence. In this technology, the genetic information is recovered directly, based on the binding of the functional protein to its target. Based on this technology, numerous methods have been developed and validated. These methods can be divided into two categories: (i) cell-based methods such as phage display [6] and cell-surface display [7,8], as well as (ii) cell-free methods such as ribosome display [9,10,11] and mRNA display [12].

Here in this review, we discuss the ribosome display technology and the applications of this in vitro system in biomedical and agricultural fields for the generation of recombinant scFv antibodies for disease diagnostics and control.

## 2. In Vitro Ribosome Display

Ribosome display technology (RDT) is a potent in vitro, cell-free system that overcomes many limitations of cell-based methods by producing in vitro protein–mRNA complexes. There are several advantages of ribosome display compared to cell-based methods. First, the method is more efficient in the screening of large libraries without compromising the library size by transformation efficiency, selecting high-affinity combining sites, and eukaryotic cell-free systems, which are capable of post-translational modifications. Furthermore, it is quick and efficient as no cell culture is involved [13,14]. On the other hand, a noted limitation in ribosome display is the accessible, functional ribosome levels in the reaction for the library, which depends on the library size.

RDT produces stable protein (antibody)–ribosome–mRNA (PRM) complexes to link individual antibody fragments to their corresponding mRNA [15]. The PRM complexes are formed through the deletion of the terminal stop codon from the mRNA, which causes stalling of the translating ribosome at the end of mRNA with the nascent polypeptide not released. The protein–mRNA linkage allows the simultaneous isolation of the mRNA and desirable proteins (antibodies) through an affinity for an immobilized ligand. The protein–mRNA complex that binds tightly to the ligand is subjected to in situ reverse transcription-PCR (RT-PCR) to recover the DNA encoding protein sequence [16] and amplified in a PCR reaction to generate a template for further manipulation and protein expression or panned for 3–5 additional cycles to obtain antibody leads. RDT allows the screening of libraries, with up to 10^12–15^-member antibody libraries in a single reaction [17].

Ribosome display has been extensively used in eukaryotic and prokaryotic translation systems. This was first demonstrated through a selection of peptide ligands using an *E. coli* extract by Mattheakis et al. [9,18]. This group showed the selection of peptide ligands that are similar to known peptide epitopes of a given antibody, using the antibody as a selection substrate. The peptide ligands of high-affinity were bound to the prostate-specific antigen and were identified through polysome selection from peptide libraries using a wheat germ extract translation system [19]. The selection of functional antibody fragments was reported using an *E. coli* translation system designed to increase the yield of ternary complexes and to allow disulfide bond formation [10]. This experimental set up was used to select antibodies from a murine library, and it was shown that the maturation of affinity occurs during the selection process. This is due to the combined effect of PCR errors and selection. An scFv fragment with a dissociation constant of about 10^−11^ M was obtained [20]. Specific antibody enrichment from mixed populations using rabbit reticulocyte extracts has also been demonstrated [10,21]. There was another study where scFv–ribosome–mRNA complexes were produced using a rabbit reticulocyte lysate system. This was then panned against the terminal protein (TP)-peptide of hepatitis B virus (HBV) DNA polymerase [22]. He and Taussig [14,16,23] also described the step-by-step procedure to perform eukaryotic ribosome display methodology. This has the distinctive feature of an in-situ RT-PCR procedure for DNA recovery from ribosome-bound mRNA. Another group [24] reported that a pseudoknot (originating from the genomic RNA of infectious bronchitis virus (IBV), a member of the positive-stranded coronavirus group) improves the selection efficiency in eukaryotic rabbit reticulocyte ribosome display. Qi et al. [25] selected antisulfadimidine-specific scFvs from a hybridoma cell through eukaryotic ribosome display. Kastelic and He [26] described the ribosome display of antibodies through the use of a eukaryotic rabbit reticulocyte system with an in-situ single-primer DNA recovery method. Edwards and He [27] also described the use of the eukaryotic rabbit reticulocyte ribosome display method to isolate variants of V(H) antibody fragments with improved affinities. Douthwaite [28] developed an optimized methodology for the use of rabbit reticulocyte lysate for ribosome display selections. Tang et al. [29] validated a novel in-vitro method for the rapid generation of human scFv monoclonal antibodies against recombinant gp120^K530^ from patient libraries using eukaryotic ribosome display.

### 2.1. Selection of Antibodies by Panning

Panning (also called biopanning) or affinity enrichment is a technique to isolate antibody fragments from a diverse antibody library based on their binding affinity to a given target [30]. A typical selection cycle is illustrated in Figure 1. The antigen of interest is immobilized on a solid surface such as nitrocellulose [31,32], magnetic beads [33], column matrices [34], plastic surfaces like polystyrene tubes [35], or 96-well microtiter plates [36]. The conformational integrity of antigens during the immobilization is critical to obtain functionally specific antibodies. Some antibodies that are selected against an adsorbed antigen may not be able to recognize the native form of the antigen [37]. Indirect antigen immobilization using antigen-specific capture antibodies may avoid this problem. 

PRM complexes are first incubated with immobilized antigens. Unbound antibody complexes are then removed by thorough washing (Figure 1). The bound PRM complexes can then be subjected to in-situ RT-PCR to recover the DNA encoding protein sequence. Because the binding of non-specific antibody fragment limits the enrichment achieved per cycle, usually 3–5 panning rounds are necessary to select specifically antibody fragments in practice. In the end, individual antibody clones can be tested by monoclonal ELISA. Afterwards, these individual binders can be sequenced and further biochemically characterized [38,39,40,41]. This panning process can also be performed in a high-throughput manner [42,43]. Because the gene sequence of the binder is available, the antibody—depending on the desired application—can be reconverted into different antibody formats (e.g., scFv-Fc fusion or IgG) and produced in different production hosts [44,45]. Affinity, but also the stability of the antibodies selected by ribosome display, can be increased by additional in-vitro affinity maturation steps [46,47,48].

### 2.2. Affinity Maturation and Modification of Ribosome Display Antibodies

The formation of a stable antibody complex and the mRNA that encodes it forms the foundation for the most developed forms of in-vitro display [49]. Then, the amplification of the mRNAs from selected complexes is performed. The field of antibody affinity maturation represents the most successful application of ribosome display [46,50,51,52,53,54]. This display system, with its built-in affinity maturation feature caused by the error-prone DNA shuffling or site-directed mutagenesis process of reverse transcriptase and amplification, enables efficient maturation of picomolar antibody concentrations [20,27,52,55,56,57,58,59,60,61,62]. With ribosome display systems employing such strategies, improvements greater than 1000-fold in potency within 6 months have been achieved for antibodies derived from phage display or from immunized animals [63]. Hanes et al. have integrated ribosome display with error-prone PCR and obtained scFvs that possess equilibrium dissociation constants about 82 pM. An increase in affinity that was almost 40-fold, when compared to progenitor clones, was caused by the point mutations introduced [50]. However, error-prone PCR, coupled with gene shuffling at later cycles in another study, gave more populations of variants with higher affinity, unlike using only error-prone PCR [64]. Recombinant antibody technology has seen new advances like the automation of high-throughput technologies and maturation of selection platforms [65], and these, when merged with the high-level process of affinity maturation, could facilitate the generation of antibodies for research and diagnostic applications.

### 2.3. Ribosome Display Antibody Gene Libraries 

Selection and affinity maturation of complex scFv antibody libraries were tested foremost using ribosome display [20,66], first from a library from immunized mice, and, subsequently, from a synthetic library [50]. Then, only the general binding protein scaffold was available, with high diversity for the recreated synthetic repertoire of the antibodies. The folding of scFvs in an in-vitro translation system has to be proportionate to their oxidative folding (usually antibody domains are required to fold correctly with intradomain V_H_ and V_L_ disulfide bonds), and, as such, this reaction needs to be catalyzed [67]. In addition, some antibodies tend to aggregate, and this is the reason why these are enriched over fewer rounds than for some other scaffolds, where their robust in-vitro folding disallows aggregation. Several publications exist on the use of phage display instead of ribosome display in the selection from native antibody libraries. Filamentous phage display [68] performs satisfactorily with secreted proteins like scFv [69], and there exists the alternative to merge the two methods, instead of using the selection and affinity maturation together in one procedure, as in the ribosome display selection from native or synthetic libraries [46,50,70,71]. In comparison to immune libraries, native, semi-synthetic, and synthetic libraries are referred to as “single-pot” libraries, depending on their abilities to isolate antibodies against antigens of interest. From the analysis of ribosome display selection from the totally synthetic antibody library HuCAL [72,73], one can infer that the selection is not exhaustive, and the result is determined by the existence of random mutations; selections performed again and again against the same target yielded several frameworks that were dominant in the different selections. This observation implies that an initial beneficial mutation may yield many (subsequently mutated) progeny of a given clone, but then, in the following selection experiment on the same target, another framework combination may have developed such a beneficial mutation.

Combinatorial libraries of a novel class of small proteins, termed “Designed Ankyrin Repeat Proteins” (DARPins) [54,74,75,76,77,78,79,80,81,82,83], were produced to serve as alternatives to antibodies since they could be robustly engineered. The proteins lack cysteine, can be expressed in high levels in a soluble form in the cytoplasm of *E. coli*, and possess great stability, exhibiting robust folding and withstanding aggregation [84,85]. Because the advantageous biophysical properties and binders with high affinity are realized at high frequency, the direct selection of binders from the richly diversified library performed excellently with DARPins [82]. Very recently, Schilling et al. [77] developed LoopDARPins, a next-generation of DARPins, with improved epitope-binding properties. Thus, binders have been isolated outrightly by ribosome display [75,86,87,88,89,90,91,92] against several targets, containing hard-to-get ones like detergent-solubilized GPCRs [93] or DNA conformers [94]. Binders based on the camelid VHH domains, having micromolar affinity, were obtained by ribosome display from a naive library [95], and possessed nanomolar affinity from an immunized llama [96]. At present, single-pot antibody libraries with a theoretical diversity of up to 10^15^ independent clones have been produced and employed for the isolation of antibodies for research, diagnostic, and therapeutic purposes [17].

## 3. Ribosome Display Technology in Disease Diagnostics and Control

Production of antibodies, especially scFvs, has been dramatically accelerated by in-vitro selection systems such as ribosome display technology. Within the past two decades, antibodies have gradually become very essential tools in the fields of biological sciences, agriculture, and medicine for basic research, disease diagnostics, and therapy. 

### 3.1. Human Infectious Diseases

A human infectious disease of major importance is Ebola virus disease (EVD) or Ebola hemorrhagic fever. The disease was first identified in 1976 during two different outbreaks in Nzara, South Sudan, and Yambuku, Democratic Republic of Congo [97]. The most common form of this disease is caused by the Ebola virus (EBOV), believed to be transmitted by fruit bats, and can also be transmitted from infected persons to other uninfected people via direct contact with body fluids [97]. Symptoms of the disease usually start about two to five days postvirus contraction and include fever, headaches, muscular pain, and sore throat, with diarrhea, vomiting, rashes, internal and external bleeding usually following these earlier symptoms [97]. Other diseases such as cholera, typhoid, and malaria do present symptoms similar to EBOV infection. There is usually a high risk of death from the disease. EVD occurs from time to time in subSahara Africa, but the world witnessed the largest EBOV disease outbreak (the West Africa epidemic) between 2013 and 2016 that was responsible for 11,323 deaths from about 28,646 cases [97]. Other lesser outbreaks have occurred subsequently [97], and great efforts have been made to improve the diagnosis and control of this disease. In line with these efforts to combat EVD, we have employed cell-free ribosome display technology to develop a panel of single-chain antibodies against virion surface epitopes of the Ebola virus that was able to detect not only the different known species of ebolaviruses but also the related Marburg virus (MARV) [98]. Besides EBOV, which many studies have basically centered on, monoclonal antibodies are rarer for other ebolavirus species or other pathogenic filoviruses such as Sudan (SUDV), Bundibugyo (BDBV), Tai Forest (TAFV), Marburg (MARV), and Marburgvirus Ravn (RAVV) viruses. This situation negatively affects antibody-based diagnostics against these pathogenic species [98]. The broadly cross-reactive scFv antibodies that we have generated have high diagnostic potentials for all species of ebolaviruses, as well as for MARV (Figure 2).

Another infectious disease that has recently become of importance is the Zika virus disease (ZVD) caused by the Zika virus (ZIKV) [99]. ZIKV is a flavivirus first identified in monkeys in 1947 and later in humans in 1952 in Uganda, and the disease is spread by *Aedes* mosquitoes, especially *Aedes aegypti* and *A. albopictus,* which are daytime biting species [99,100]. The disease occurs in Africa, Asia, the Pacific, and the Americas. Zika infection usually presents no symptoms or mild ones such as fever, headache, rash, red eyes, and joint pain [100]. However, the disease is particularly dangerous because it can be passed on during pregnancy from mother to an unborn fetus, where it can cause microcephaly, other congenital abnormalities, and pregnancy complications [100]. An outbreak of ZVD in Brazil in 2015 raised great concern regarding its association with microcephaly, and thereafter, several outbreaks started to occur in different parts of the world, such as in the Americas and in Africa [100]. There are no approved vaccines yet for this disease and diagnostics were, until recently, quite limited. To bridge these gaps, our laboratory utilized ribosome display to generate high-affinity scFv antibodies that have specificity to ZIKV envelope proteins [101]. In addition to their high-affinity binding and specificity, these scFvs were also able to neutralize live ZIKV and inhibit infectivity [101]. These single-chain antibodies have great potential and could serve as diagnostics or treatments of ZIKV infections (Figure 3).

Unlike Zika and Ebola diseases, which have become known to mankind within the last century, tuberculosis (TB) is an older infectious disease known since hundreds of years ago. TB is caused by *Mycobacterium tuberculosis* and was responsible for about 10 million infections and 1.5 million deaths worldwide in 2018 [102]. Diagnosis for this disease has challenges as one of the two major methods is not very reliable for immune-compromised or previously vaccinated people, while the second method is expensive and can only be performed by expert staff [103,104]. This necessitates a cheaper and simpler diagnostic method. To this end, Ahangarzadeh et al. [104] utilized ribosome display technology to generate scFvs that are specific against the early secretory antigenic target (ESAT-6) antigen of *M. tuberculosis*. It is expected that the scFv against ESAT-6 will facilitate the development of a simple, fast, and cheap diagnostic kit for TB.

The aforementioned successes in the generation of scFvs specific to desired targets strongly suggest that ribosome display technology could further be applied to yield scFvs for diagnosis and possible treatment of other infectious human diseases that, to date, lack proper diagnostics or control.

Recently, the world witnessed an outbreak of a pandemic, the coronavirus disease 2019 (COVID-19) [105]. This disease is caused by severe acute respiratory syndrome coronavirus 2 (SARS-CoV-2). COVID-19 continues to spread and has caused over 283,000 deaths worldwide [106], while there is currently no specific treatment or vaccine against it [105]. Monoclonal and single-chain antibodies have previously been developed against other coronaviruses, SARS-CoV and MERS-CoV, that cause severe acute respiratory syndrome (SARS) and the Middle East respiratory syndrome (MERS), respectively [107,108,109,110]. Ribosome display technology could possibly be applied to target SARS-CoV-2 to produce neutralizing scFvs. In this regard, the surface spike proteins would be an ideal target to produce scFvs that might potentially have neutralizing properties. The spike proteins play vital roles in virus–cell membrane fusion and subsequent viral entry and have been a primary target in previous studies against other coronaviruses [111]. Neutralizing scFvs against SARS-CoV-2 would help in the development of diagnostics and treatment for COVID-19. 

### 3.2. Cancer

Cancer remains a dreaded and major killer illness worldwide. An effective cure for cancer is still farfetched despite years of concerted research efforts. Radio and chemotherapy are the main strategies employed to mitigate cancer illnesses, even though these strategies have serious side-effects. Antibodies have the potential for cancer or tumor treatment since they can bind specifically to target antigens on cancer cell surfaces. However, a major challenge for the use of antibodies in cancer treatment is the somewhat extensive screening required to obtain antibodies that have high specificity and affinity against the desired target antigens. Ribosome display is a powerful and ideal in-vitro tool that can perform such screening tasks for highly specific and high-affinity single-chain antibodies [54,112]. Reasoning along this line, Huang et al. [113] applied ribosome display to perform large-scale screenings of scFvs against tumor cells. scFvs with high affinity for cancer stem cells were obtained in their study, and the activities of these scFvs could hinder the growth of cancer cells in vitro and in vivo. These new antibodies could usher in a new way for cancer treatment that is devoid of undesired side-effects.

### 3.3. Acquired Immunodeficiency Syndrome (AIDS)

The human immunodeficiency virus (HIV) is a virus that can attack the human immune system and advance to cause the disease called AIDS if left untreated [114]. The virus is thought to have originated from a similar version in chimpanzees in Central Africa, known as the simian immunodeficiency virus (SIV), which may have crossed into humans [114]. As at the end of 2018, there were about 37.9 million people living with HIV worldwide [115]. No cure exists for HIV yet, but the disease can be managed through antiretroviral therapy (ART) [114]. New developments in antibody research could provide a game-changing breakthrough in the fight against this disease. Monoclonal antibodies that have broad neutralizing effects towards HIV have been identified and characterized [116], with the aim of determining epitopes that could be helpful in designing mimetic structures to induce antibodies with broad protection against the virus. More broadly neutralizing antibodies will, therefore, be needed in this approach towards developing good vaccines against HIV [117,118]. However, most of the methods applied were not well suited for the task as they have limitations, such as being labor-intensive, time-consuming, the diversity of library repertoire they can screen being limited, in addition to the relatively high costs involved. Ribosome display offers a cheaper and faster cell-free strategy to accomplish the goal of screening and selecting neutralizing antibodies. Tang et al. [29] demonstrated this possibility when they utilized ribosome display to rapidly produce monoclonal antibodies in vitro by directly screening single-chain antibody repertoires that were derived from peripheral blood mononuclear cells (PBMCs) of HIV patients. Going forward, this display technology can potentially lead to the generation of diverse antibodies that may facilitate the development of an effective vaccine against HIV.

### 3.4. Plant Disease: Pierce’s Disease

Pierce’s disease (PD) is currently a problem facing the Californian grape industry. PD is caused by *Xylella fastidiosa*, a Gram-negative bacterium that is limited to the xylem of the plant [119]. This disease is transmitted by sap-sucking insects such as the glassy-winged sharpshooter (GWSS) that feeds on xylem vessels and passes the bacterium it picks up during feeding from infected plants to uninfected ones [120,121,122]. *X. fastidiosa* normally attaches to the interior of the foregut of the insect and then gets transmitted from one plant to another [121,122,123]. From the inoculation site, *X. fastidiosa* multiplies and spreads to colonize the xylem, blocking the water transport network, causing scorch- like symptoms. GWSS has become established and prevalent in California, and PD is a threat to grape production. An approach to control PD is to inhibit the transmission of the pathogen *X. fastidiosa* by the invasive GWSS insect vector. A better understanding of the complex interactions between the plants, pathogens, and insects [124] and the molecular mechanisms involved may provide important information to aid the fight to prevent or reduce pathogen transmission. However, very little is known about the basis of these complex interactions.

Release of the *X. fastidiosa* genome sequence [125,126] has enabled the study of the surface proteins of *X. fastidiosa,* which may furnish targets for interventions against PD. Predictions and exploration could possibly yield surface-exposed components that may have roles in the pathogen virulence or involved in the formation or attachment of biofilms in the vector. Recently the expression of afimbrial and fimbrial proteins of *X. fastidiosa* during biofilm formation was investigated. It was found that these proteins show different patterns of distribution in the xylem during biofilm formation [127]. Furthermore, haemagglutinin adhesion and MopB, an outer membrane protein, have been studied in *X. fastidiosa* [120,128,129,130]. While the role of the protein (MopB) is not well known, it is well established that the outer membrane proteins (OMPs) in Gram-negative bacteria play vital roles such as (1) keeping the structural integrity of the outer membrane (OM), (2) recognition proteins, (3) transportation, (4) membrane pores, (5) membrane-bound enzymes or components of signal cascades [131,132,133,134], (6) stress resistance (implicated are *Escherichia coli* OmpA and OprF in *Pseudomonas aeruginosa*) [135,136,137], (7) pathogenesis (for example, OmpA in *Escherichia coli* and OspC in *Borrelia burgdorferi*) [134,138,139], and (8) agglutination. Polyclonal antibodies and lectins can also be used to probe the function of targets displayed on the pathogen cell surface [140]. Developing single-chain antibodies (scFvs) against suitable surface protein targets on *X. fastidiosa* could be a key strategy to hinder bacterial attachment and to stop PD. The production of scFv antibodies is a potential avenue for the generation of anti-*Xylella* factors. Using a phage antibody library, Lampe et al. [141] attempted to screen for scFvs against *X. fastidiosa*’s outer protein coat [141,142]. Recently, Azizi et al. [143] demonstrated a simple and robust method for the generation of panels of recombinant scFvs using a eukaryotic rabbit reticulate system against the surface-exposed element or outer membrane protein, MopB, of *X. fastidiosa* from in-vitro combinatorial antibody ribosome display libraries. The in-vitro anti-*X. fastidiosa* scFv libraries produced in the study and the strategy for the preparation of recombinant putative membrane proteins provide approaches for the rapid discovery of additional scFvs against surface components involved in aggregation [144] and/or motility [145,146,147]. The anti-MopB or other potential anti-*X. fastidiosa* scFv molecules could be useful in developing diagnostics for surveillance of the pathogen and could be coupled with fluorophores, as recently described [148,149]. Moreover, recombinant antibodies against MopB and other abundant surface-exposed molecules on *X. fastidiosa* could be engineered to agglutinate the bacteria and be introduced into the GWSS via paratransgenic organisms such as engineered *Pantoea agglomerans, Metarhizium spp* [150], or *Beauvaria bassiana* [151], or an avirulent strain of *Xylella* itself [152], providing new platforms to investigate the control of PD. Our laboratory is refining this technology employed by Azizi et al. [143] to develop panels of scFvs against other surface epitopes of the plant pathogen *X. fastidiosa*. Blocking of the surface epitopes with antibodies may curb the transmission of the pathogen. Therefore, these scFv antibodies may potentially be used in the future for diagnosis and (or) disease control of PD.

## 4. Future Perspectives

Ribosome display has proven to be a robust procedure, used now in academic and industrial laboratories, which comes rather close to experimental protein evolution in test tubes. Undoubtedly, the procedure will be further improved and applied to many new targets and selection goals. Together with the introduction of new technologies like next-generation sequencing, robotics, and nanotechnology, high-throughput screening of ribosome display libraries for rapid antibody generation is now a reality. Moreover, new molecular-based techniques for library generation and panning strategies will set the tone for the constant improvement of ribosome display in antibody generation for human infectious diseases, plant diseases, and other diagnostics. Without a doubt, this technology will continue to evolve and play a bigger role in the coming decade within research, therapeutic, and diagnostic markets. 

## Figures and Tables

**Figure 1 antibodies-09-00028-f001:**
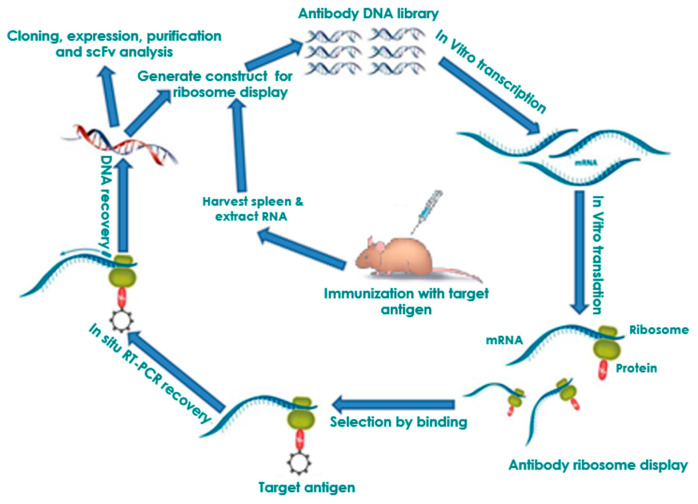
Principle of in vitro ribosome display. DNA library is first amplified by PCR as a T7 promoter, ribosome binding site, the gene, the spacer, and no stop codon. The amplified DNA library is used in an in-vitro coupled transcription/translation to form mRNA, the related protein, and the ribosome complex (PRM complex). The PRM complexes are affinity selected from the transcription/translation mixture by binding of the immobilized antigen. The bound PRM complexes can then be subjected to in-situ RT-PCR to recover the DNA encoding protein sequence and PCR-amplified for an additional selection cycle or postselection analysis.

**Figure 2 antibodies-09-00028-f002:**
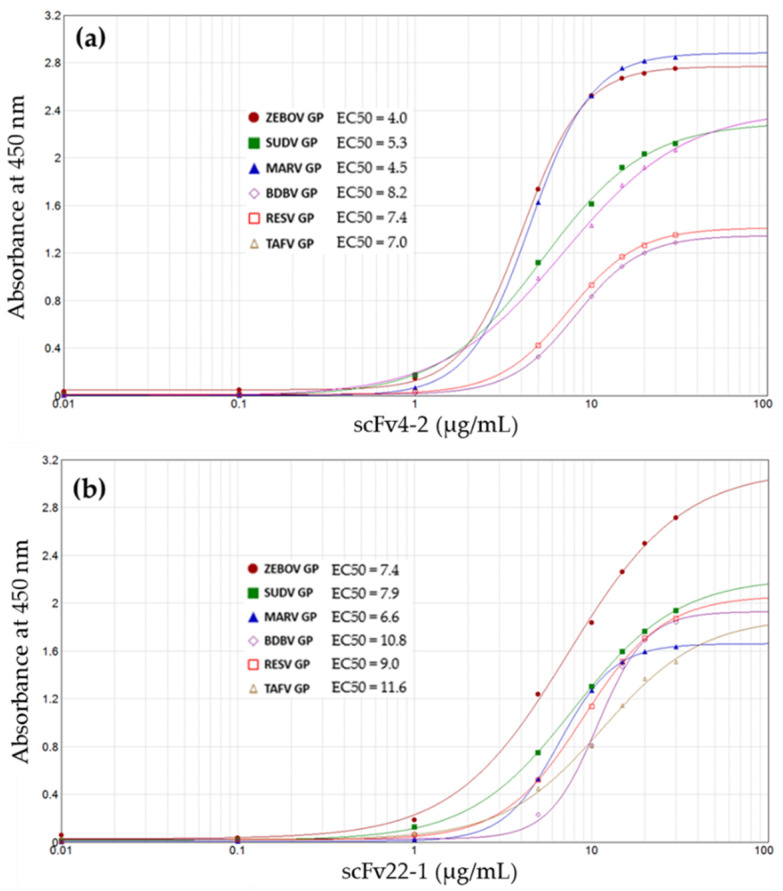
Pan-filovirus single-chain antibody fragments (scFvs) generated by advanced ribosomal display (adapted from Kunamneni et al. [98]). (**a**,**b**) ELISA dose–response curves show specific binding of pan-filovirus scFv4-2 and scFv22-1. Differential binding of these scFvs to glycoprotein (GP) from five ebolaviruses (Zaire (ZEBOV), Sudan (SUDV), Tai Forest (TAFV), Bundibugyo (BDBV), and as well as the non-pathogenic Reston (RESTV)) and one Marburg virus (MARV) was determined by a 4-parameter logistic ELISA curve [98].

**Figure 3 antibodies-09-00028-f003:**
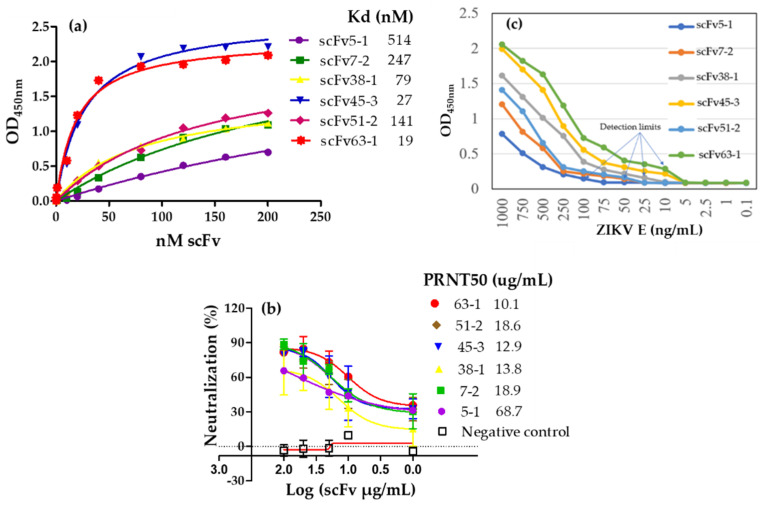
Zika envelope binding and neutralizing analysis of scFvs (adapted from Kunamneni et al. [101]). (**a**) Binding curve of scFvs to Zika E by ELISA. (**b**) Plaque reduction of ZIKV PRVABC59 with 6 scFv antibodies (5-1, 7-2, 38-1, 39-2, 45-3 and 69-3). These data show the feasibility of generating neutralizing scFvs by ribosomal display [101]. (**c**) Antigen detection limits of the scFv were determined by antigen titration ELISA. About 10 ng/mL were detected by scFv45-3 and scFv63-1, ~25 ng/mL by scFv38-1, ~50 ng/mL by scFv7-2 and scFv51-2, and ~100 ng/mL by scFv5-1. scFvs against Zika E protein show that sub-nanomolar quantities of antigen can be detected with this method, suggesting that this approach can attain adequate sensitivity for diagnostic purposes.

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
