# Peer review of "Ribosome Display Technology: Applications in Disease Diagnosis and Control"

_2073-4468, 2020, doi:10.3390/antib9030028_

Round 1
Reviewer 1 Report
The manuscript will benefit more from the usage of figures with higher resolutions.
It is not clear if Fig. 1 has been obtained from another article.
Line 83 - representation of reference 22.
From Line 88 to Line 97 - It is not clear which novelties or improvements have been brought to the literature in the cited articles.
Line 115 - Citation style
Figure 3 is not clear.
Author Response
Comments and Suggestions for Authors
The manuscript will benefit more from the usage of figures with higher resolutions.
Response: Figures with higher resolution have been put in the manuscript.
It is not clear if Fig. 1 has been obtained from another article.
Response: New figure has been included for Figure 1 in the manuscript.
Line 83 - representation of reference 22.
Response: Yes, line 83 is a representation of reference 22 and has been formatted accordingly.
From Line 88 to Line 97 - It is not clear which novelties or improvements have been brought to the literature in the cited articles.
Response: We have modified the text from Line 88 to Line 97 as per reviewer’s valuable suggestion.
Line 115 - Citation style
Response: The citation in line 115 has been corrected to journal citation style.
Figure 3 is not clear.
Response: A clearer version of Fig. 3 has been included in the manuscript.
Reviewer 2 Report
The paper “Ribosome Display Technology: Applications in Disease Diagnosis and Control” attempts to review the latest developments in the area of ribosome display technology. The subject lacks up-to-date reviews, so this could be a very good opportunity to do so. However, in the way it is, I don’t see how this objective can be achieved.
For a review, the paper touches too lightly on the description of the panning process and on the use of the technology for affinity maturation (despite the enormous amount of references). It doesn’t bring anything new, or better, than what one can find on another recently published article treating exactly the same subject (“Ribosome Display: A Potent Display Technology used for Selecting and Evolving Specific Binders with Desired Properties”, PMID 30406440).
The part that could be interesting and bring real novelty, on “Disease Diagnostics and Control”, is sadly lacking, basically citing work by the author’s group. Ribosome display is not my field, but a quick search shows that there’s more to be included in that area than what has been included in the manuscript.
Moreover, it perplexes me that Figures 2 and 3 are copies from their former papers, with minimal or no modifications (and not clearly stating so).
Finally, the writing is poor and containing grammar and form mistakes. Examples (not exhaustive) of sentences lacking sense are:
“Using such approaches in ribosome display systems lead antibodies derived from phage display or from immunized animals have been improved >1000-fold in potency within 6 months [63].”
“In another study, the combination of error-prone PCR and gene shuffling in subsequent cycles yielded an increased population of affinity-improved variants, and the highest affinity clone, as compared to error-prone PCR only [64].”
“At that time, the only general binding protein scaffold was available with great diversity for the recreated synthetic repertoire of the antibodies.”
“and this is the reason why these are enriched over fewer rounds than for some other scaffolds that their robust in vitro folding.”
“The proteins contain no cysteine, can be expressed in soluble form with very high levels of the cytoplasm of E. coli”
“monoclonal antibodies are rarely available for different other Ebola species”
Author Response
Comments and Suggestions for Authors
The paper “Ribosome Display Technology: Applications in Disease Diagnosis and Control” attempts to review the latest developments in the area of ribosome display technology. The subject lacks up-to-date reviews, so this could be a very good opportunity to do so. However, in the way it is, I don’t see how this objective can be achieved.
For a review, the paper touches too lightly on the description of the panning process and on the use of the technology for affinity maturation (despite the enormous amount of references). It doesn’t bring anything new, or better, than what one can find on another recently published article treating exactly the same subject (“Ribosome Display: A Potent Display Technology used for Selecting and Evolving Specific Binders with Desired Properties”, PMID 30406440).
Response: Unlike the article stated, the main focus of our article is on the application of the ribosome display technology to produce antibodies for disease diagnostics and control.
The part that could be interesting and bring real novelty, on “Disease Diagnostics and Control”, is sadly lacking, basically citing work by the author’s group. Ribosome display is not my field, but a quick search shows that there’s more to be included in that area than what has been included in the manuscript.
Response: Works by others on ribosome display for diagnostics and control of diseases have been added, particularly those on cancer treatment and control of respiratory infectious diseases due to coronaviruses.
Moreover, it perplexes me that Figures 2 and 3 are copies from their former papers, with minimal or no modifications (and not clearly stating so).
Response: The citation numbers in the legends for Figures 2 and 3 had indicated our publications (references [99] and [102]) that these Figures were taken from. However, we have also now clearly stated in the figure legends that Figures 2 and 3 were from our former publications.
Finally, the writing is poor and containing grammar and form mistakes. Examples (not exhaustive) of sentences lacking sense are:
“Using such approaches in ribosome display systems lead antibodies derived from phage display or from immunized animals have been improved >1000-fold in potency within 6 months [63].”
“In another study, the combination of error-prone PCR and gene shuffling in subsequent cycles yielded an increased population of affinity-improved variants, and the highest affinity clone, as compared to error-prone PCR only [64].”
“At that time, the only general binding protein scaffold was available with great diversity for the recreated synthetic repertoire of the antibodies.”
“and this is the reason why these are enriched over fewer rounds than for some other scaffolds that their robust in vitro folding.”
“The proteins contain no cysteine, can be expressed in soluble form with very high levels of the cytoplasm of E. coli”
“monoclonal antibodies are rarely available for different other Ebola species”
Response: Corrections have been made to the above sentences and others. The correct versions are below:
Line 154: “Using such approaches in ribosome display systems, antibodies derived from phage display or from immunized animals have been improved >1000-fold in potency within 6 months [63].”
Line 159-161: “In another study, the combination of error-prone PCR and gene shuffling in subsequent cycles yielded an increased population of affinity-improved variants, and the highest affinity scFv, as compared to using error-prone PCR only [64].”
Line 169: “initially from a library from immunized mice, and later from a synthetic library [50]."
Line 170 “At that time, only the general binding protein scaffold was available with great diversity for the recreated synthetic repertoire of the antibodies.”
Line 174-176: “In addition, some antibodies tend to aggregate, and this is the reason why these are enriched over fewer rounds than for some other scaffolds that their robust in vitro folding disallows aggregation.”
Line 193-194: “The proteins contain no cysteine, can be expressed in soluble form with very high levels in the cytoplasm of E. coli, are very stable, exhibiting robust folding.”
Line 238: “monoclonal antibodies are rare for different other Ebola species”
Reviewer 3 Report
This manuscript reviewed the ribosome display technology, affinity maturation strategies and applications in generation and modifications of recombinant antibodies for disease diagnostics and control. It is well written and nice to read. The only suggestion I have is to add a few lines or paragraphs to describe published work or perspective of possible application to target Covid-19 virus.
Author Response
Comments and Suggestions for Authors
This manuscript reviewed the ribosome display technology, affinity maturation strategies and applications in generation and modifications of recombinant antibodies for disease diagnostics and control. It is well written and nice to read. The only suggestion I have is to add a few lines or paragraphs to describe published work or perspective of possible application to target Covid-19 virus.
Response: A paragraph on the perspective of developing single chain antibodies against COVID-19 using ribosome display technology has been added.
“Recently, the world witnessed an outbreak of a pandemic, the coronavirus disease 2019 (COVID-19) [WHO 2020a]. This disease is caused by the severe acute respiratory syndrome coronavirus 2 (SARS-CoV-2). COVID-19 continues to spread and has caused over 283,000 deaths worldwide [WHO, 2020b], while there is currently no specific treatment or vaccine against it [WHO 2020a]. Monoclonal and single chain antibodies have previously been developed against other coronaviruses, SARS-CoV and MERS-CoV, that cause the severe acute respiratory syndrome (SARS) and the Middle East respiratory syndrome (MERS) respectively [Liu et al. 2005; Leung et al. 2008; Zhu et al. 2007; Zhou et al. 2019]. However, no SARS-CoV-2-specific neutralizing antibodies have been reported so far [Jiang et al. 2020]. The ribosome display technology could possibly be applied to target SARS-CoV-2 to produce neutralizing scFvs. In this regards, the surface spike proteins would be an ideal target to produce scFvs that might potentially have neutralizing properties. The spike proteins play vital roles in virus-cell membrane fusion and subsequent viral entry, and have been a primary target in previous studies against other coronaviruses [Jiang et al 2020]. Neutralizing scFvs against SARS-CoV-2 would help in the development of diagnostics and treatment for COVID-19.”
Round 2
Reviewer 2 Report
Dear authors,
Thanks for this revised version of your paper entitled "Ribosome Display Technology: Applications in Disease Diagnosis and Control".
I think the paper was improved by adding the extra sections, but issues with writing still remain. It is clearly (by the presence of very simple grammar mistakes and typos) that the paper hasn't been properly proof-read once. The manuscript does need to be revised by a native speaker. Particularly the abstract should be re-written. The ideas don't flow in a coherent way, and some sentences make no sense (like "The peculiarity of direct generation of whole proteins", "to introduce evolution of antibodies"). And if the focus is diagnosis and control, say it (what it says now still is "ribosome display technology, approaches for affinity maturation for this in vitro system, and its far-reaching applications in the biomedical and agricultural fields in the generation of recombinant scFvs for disease diagnostics and control").
It's missing a small conclusion for part 2 (what's the summary of the current state of the technology?)
Item 5 is a bit confuse as it is. Instead of giving a run down of the different binder formats, and for which of them RDT has been employed, the authors present the problems of scFv, then say the techniques can be combined (without further explanation), then discuss the output of HuCAL libraries (7 lines!). Very very confuse.
Fig. 1, Word correction underline (scFv) is still visible
I still think figures 2 and 3 shouldn't be on the format they are. One cannot simply copy and paste data already published, even if the authors are the ones that made them (most of the time, rights of published papers are owned by the journal, and permission to reproduce needs to be granted). Re-do them, present in a nicer, condensed way. You don't need to show data (this is already published), you just want to give examples.
As for English, again, I'm just listing SOME examples of errors. There are more. Do not be stuck with these when correcting. Use a professional service, please:
- "one of the most widespread rAb format", should be FORMATS
- "Single-chain (scFv) antibodies are one of the most widespread rAb format as they have been engineered for many applications", this sentence is empty and it doesn't mean anything
- Page 9, sentence lines 52-55 poorly written. And next sentence should be "Furthermore, THEY ARE"
- Sentence page 9, lines 60-62 repeated.
- "subsequent cycles of about 3-5 to obtain antibody leads", sentence makes no sense
- "This experimental set up was been used to select", wrong grammar
- "Specific antibody enrichment from mixed populations using rabbit reticulocyte extracts have also been demonstrated", it should be HAS
- "The formation of a stable antibody complex and the mRNA that encodes it form the foundation for the most developed forms of in vitro display", should be FORMS. And "most developed" is a very loose concept
- "The field of antibody affinity maturation represents the most successful applications of ribosome display", should be APPLICATION
- "This display system .... enable efficient maturation", should be ENABLES
- "improvements >1000-fold in...", missing preposition
- "Yet, error-prone PCR coupled with gene shuffling at later cycles in another study gave more population of variants with higher affinity, and the highest affinity clone, unlike using only error-prone PCR", extremely confuse sentence
- "Selection and affinity maturation of complex scFv antibody library", should be LIBRARIES
- "Then, only the general binding protein scaffold was available with high diversity for the recreated synthetic repertoire of the antibodies", I can't understand this sentence. THEN indicates temporality?
- "and aa such this reaction needs to be catalyzed", should be AS
- "The analysis of ribosome display selection from the totally synthetic antibody library HuCAL one can infer that the selection", wrong grammar
- " were produced that can stand in as options to antibodies"wrong grammar, and also wrong concept
- "The proteins lacks cysteine", should be LACK
- "Production of antibodies, especially scFvs, have been dramatically", it's HAS
- " In recent times, antibodies have become very essential tools in the fields of biological sciences". Really, only in recent times? They have been used for decades!
- "are rare for different other Ebola species", you corrected the sentence, and it's still wrong
- "the single chain antibodies generated have great potential and could serve the either or the dual purpose as diagnostics or for treatment of ZIKV infections", confuse writing
- "Ribosome display is a powerful and an ideal in vitro tool that can perform such screening tasks for high specific and high affinity single chain antibodies", this sentence, in a way or another, has already been said countless times
- "There have been identifications and characterizations of monoclonal antibodies", very odd writing
- "being labor intensvie, consume much time", typo, TIME-CONSUMING is the word you looking for...
- "high costs invovled.", typo
- "Nevertheless, investigations on the interfaces between the pathogen, plant, insect interactions could reveal sites for molecular interventions that could confer resistance or reduce transmission of the pathogen.", sentence repeated with the one before
- "Predictions and exploration cold possibly", COULD
- "an outer membrane proteins", PROTEIN
- "The in vitro anti-X. fastidiosa scFv libraries produced in the study and the strategy for preparation of recombinant putative membrane proteins yield approaches for the rapid discovery of additional scFv's ..." very confuse sentence. Use only way of writing scFvs
- "this evolution technology will itself evolve play a bigger role in the coming decade", wrong grammar
Author Response
Dear authors,
Thanks for this revised version of your paper entitled "Ribosome Display Technology: Applications in Disease Diagnosis and Control".
I think the paper was improved by adding the extra sections, but issues with writing still remain. It is clearly (by the presence of very simple grammar mistakes and typos) that the paper hasn't been properly proof-read once. The manuscript does need to be revised by a native speaker. Particularly the abstract should be re-written. The ideas don't flow in a coherent way, and some sentences make no sense (like "The peculiarity of direct generation of whole proteins", "to introduce evolution of antibodies"). And if the focus is diagnosis and control, say it (what it says now still is "ribosome display technology, approaches for affinity maturation for this in vitro system, and its far-reaching applications in the biomedical and agricultural fields in the generation of recombinant scFvs for disease diagnostics and control").
Response: the abstract has been rewritten as suggested. The focus of this review on disease diagnostics and control has been clearly stated.
It's missing a small conclusion for part 2 (what's the summary of the current state of the technology?)
Response: We have included about the summary of the current state of the technology as per reviewers valuable suggestion.
Item 5 is a bit confuse as it is. Instead of giving a run down of the different binder formats, and for which of them RDT has been employed, the authors present the problems of scFv, then say the techniques can be combined (without further explanation), then discuss the output of HuCAL libraries (7 lines!). Very very confuse.
Response: We have rewritten the item 5 as per reviewers suggestion.
Fig. 1, Word correction underline (scFv) is still visible
Response: scFv is no longer visible in Fig. 1.
I still think figures 2 and 3 shouldn't be on the format they are. One cannot simply copy and paste data already published, even if the authors are the ones that made them (most of the time, rights of published papers are owned by the journal, and permission to reproduce needs to be granted). Re-do them, present in a nicer, condensed way. You don't need to show data (this is already published), you just want to give examples.
Response: We have redone the figures 2 and 3 as per reviewer’s valuable suggestion. I think, we don’t need copy right permissions for these from Journals.
As for English, again, I'm just listing SOME examples of errors. There are more. Do not be stuck with these when correcting. Use a professional service, please:
- "one of the most widespread rAb format", should be FORMATS
Response: correction effected.
- "Single-chain (scFv) antibodies are one of the most widespread rAb format as they have been engineered for many applications", this sentence is empty and it doesn't mean anything
Response: the sentence has been corrected to “Single-chain fragment variable (scFv) antibodies are one of the most widespread rAb formats and have been engineered for many applications”
- Page 9, sentence lines 52-55 poorly written. And next sentence should be "Furthermore, THEY ARE"
Response: the correction of the sentence before this sentence has been modified and makes correction of this sentence unnecessary.
- Sentence page 9, lines 60-62 repeated.
Response: the lines were not repetitions, but further explanation.
- "subsequent cycles of about 3-5 to obtain antibody leads", sentence makes no sense
Response: the sentence has been corrected to “panned for about 3-5 additional cycles to obtain antibody leads”
- "This experimental set up was been used to select", wrong grammar
Response: the sentence has been corrected to “This experimental set up was being used to select”
- "Specific antibody enrichment from mixed populations using rabbit reticulocyte extracts have also been demonstrated", it should be HAS
Response: corrected to “has”
- "The formation of a stable antibody complex and the mRNA that encodes it form the foundation for the most developed forms of in vitro display", should be FORMS. And "most developed" is a very loose concept
Response: corrected to “forms”
- "The field of antibody affinity maturation represents the most successful applications of ribosome display", should be APPLICATION
Response: corrected to application
- "This display system .... enable efficient maturation", should be ENABLES
Response: corrected to “enables”
- "improvements >1000-fold in...", missing preposition
Response: corrected to “improvements greater than 1000-fold”
- "Yet, error-prone PCR coupled with gene shuffling at later cycles in another study gave more population of variants with higher affinity, and the highest affinity clone, unlike using only error-prone PCR", extremely confuse sentence
Response: corrected to "Yet, error-prone PCR coupled with gene shuffling at later cycles in another study gave more population of variants with higher affinity, unlike using only error-prone PCR"
- "Selection and affinity maturation of complex scFv antibody library", should be LIBRARIES
Response: corrected to “libraries”
- "Then, only the general binding protein scaffold was available with high diversity for the recreated synthetic repertoire of the antibodies", I can't understand this sentence. THEN indicates temporality?
Response: yes, “then” indicates temporality. What we meant was “at that point in time”
- "and aa such this reaction needs to be catalyzed", should be AS
Response: “aa” corrected to “as”
- "The analysis of ribosome display selection from the totally synthetic antibody library HuCAL one can infer that the selection", wrong grammar
Response: sentence corrected to "From the analysis of ribosome display selection from the totally synthetic antibody library HuCAL one can infer that the selection"
- " were produced that can stand in as options to antibodies"wrong grammar, and also wrong concept
Response: corrected to “were produced that can stand in as alternatives to antibodies”
- "The proteins lacks cysteine", should be LACK
Response: corrected to “lack”
- "Production of antibodies, especially scFvs, have been dramatically", it's HAS
Response: corrected to “has”
- " In recent times, antibodies have become very essential tools in the fields of biological sciences". Really, only in recent times? They have been used for decades!
Response: the sentence has been corrected to “Within the past two decades, antibodies have become very essential tools”
- "are rare for different other Ebola species", you corrected the sentence, and it's still wrong
Response: corrected to “are rare for other Ebola species”
- "the single chain antibodies generated have great potential and could serve the either or the dual purpose as diagnostics or for treatment of ZIKV infections", confuse writing
Response: the sentence has been corrected to “the single chain antibodies generated have great potential and could serve only one or the dual purpose as diagnostics or for treatment of ZIKV infections”
- "Ribosome display is a powerful and an ideal in vitro tool that can perform such screening tasks for high specific and high affinity single chain antibodies", this sentence, in a way or another, has already been said countless times
Response: we cannot over-emphasize how good and ideal this method is for different applications.
- "There have been identifications and characterizations of monoclonal antibodies", very odd writing
Response: the sentence has been restructured to “Monoclonal antibodies that have broad neutralizing effects towards HIV have been identified and characterized [117], with the aim”
- "being labor intensvie, consume much time", typo, TIME-CONSUMING is the word you looking for...
Response: corrected to "being labor intensive, time-consuming"
- "high costs invovled.", typo
Response: corrected to “involved”
- "Nevertheless, investigations on the interfaces between the pathogen, plant, insect interactions could reveal sites for molecular interventions that could confer resistance or reduce transmission of the pathogen.", sentence repeated with the one before
Response: sentence deleted
- "Predictions and exploration cold possibly", COULD
Response: corrected to “could”
- "an outer membrane proteins", PROTEIN
Response: corrected to “outer membrane protein”
- "The in vitro anti-X. fastidiosa scFv libraries produced in the study and the strategy for preparation of recombinant putative membrane proteins yield approaches for the rapid discovery of additional scFv's ..." very confuse sentence. Use only way of writing scFvs
Response: the sentence has been corrected to “The in vitro anti-X. fastidiosa scFv libraries produced in the study and the strategy for preparation of recombinant putative membrane proteins provide approaches for the rapid discovery of additional scFvs”
- "this evolution technology will itself evolve play a bigger role in the coming decade", wrong grammar
Response: corrected to "this evolution technology will itself evolve and play a bigger role in the coming decade”